# Gluten Exorphins Promote Cell Proliferation through the Activation of Mitogenic and Pro-Survival Pathways

**DOI:** 10.3390/ijms24043912

**Published:** 2023-02-15

**Authors:** Federico Manai, Lisa Zanoletti, Giulia Morra, Samman Mansoor, Francesca Carriero, Elena Bozzola, Stella Muscianisi, Sergio Comincini

**Affiliations:** 1Department of Biology and Biotechnology “L.Spallanzani”, University of Pavia, 27100 Pavia, Italy; 2Laboratory for Mucosal Immunology, TARGID, Department of Chronic Diseases, Metabolism and Ageing, KU Leuven, 3000 Leuven, Belgium; 3SCITEC, Consiglio Nazionale delle Ricerche, 20131 Milano, Italy; 4Department of Physiology and Biophysics, Weill Cornell Medicine, New York, NY 10065, USA; 5Pediatric Unit, I.R.C.C.S. Bambino Gesù Children Hospital, 00165 Roma, Italy; 6Cell Factory and Pediatric Hematology/Oncology, Fondazione IRCCS Policlinico San Matteo, 27100 Pavia, Italy

**Keywords:** celiac disease, gluten, gluten exorphins, opioid receptors, δ-opioid receptor

## Abstract

Celiac disease (CD) is a chronic and systemic autoimmune disorder that affects preferentially the small intestine of individuals with a genetic predisposition. CD is promoted by the ingestion of gluten, a storage protein contained in the endosperm of the seeds of wheat, barley, rye, and related cereals. Once in the gastrointestinal (GI) tract, gluten is enzymatically digested with the consequent release of immunomodulatory and cytotoxic peptides, i.e., 33mer and p31-43. In the late 1970s a new group of biologically active peptides, called gluten exorphins (GEs), was discovered and characterized. In particular, these short peptides showed a morphine-like activity and high affinity for the δ-opioid receptor (DOR). The relevance of GEs in the pathogenesis of CD is still unknown. Recently, it has been proposed that GEs could contribute to asymptomatic CD, which is characterized by the absence of symptoms that are typical of this disorder. In the present work, GEs cellular and molecular effects were in vitro investigated in SUP-T1 and Caco-2 cells, also comparing viability effects with human normal primary lymphocytes. As a result, GEs treatments increased tumor cell proliferation by cell cycle and Cyclins activation as well as by induction of mitogenic and pro-survival pathways. Finally, a computational model of GEs interaction with DOR is provided. Altogether, the results might suggest a possible role of GEs in CD pathogenesis and on its associated cancer comorbidities.

## 1. Introduction

Celiac disease (CD), also referred as celiac sprue and gluten-sensitive enteropathy, is a chronic immune-mediated enteropathy affecting the small intestine of individuals with a genetic predisposition which is also characterized by extra-intestinal symptoms. This disorder is triggered by the ingestion of gluten, a mixture of storage proteins in the seeds of mature grains of wheat, barley, and rye. Particularly, the inflammatory and immune-mediated response is caused by the biological activity exerted by immunogenic and cytotoxic peptides released after the enzymatic digestion of gluten in the gastrointestinal (GI) tract [1,2]. According to the Oslo definition and subsequent clarifications [3,4], CD can be subdivided into symptomatic CD, in which it is possible to recognize different subtypes (i.e., classic CD, subclinical CD, and refractory CD) [1,5].

Gluten exorphins (GEs) are a family of small peptides (four/five amino acids) derived from the enzymatic digestion of gluten in the GI tract exerted by pancreatic elastase [6]. These peptides, described as agonist of opioid receptors (ORs), were originally identified in 1979 by Zioudrou and colleagues [7]. The role of GEs in CD pathogenesis is still unknown. Recently, it has been proposed that GEs could play a key role in the onset of the asymptomatic form of CD, by masking the deleterious effects of gluten protein on gastrointestinal lining and function [8]. GEs (classified as A4, A5, B4, B5, C5) showed affinity for ORs with different selectivity (EC_50_) for δ-opioid receptor (DOR), κ-opioid receptor (KOR) and μ-opioid receptor (MOR). Particularly, B5 is the most potent among GEs and exerts its biological activity mainly through the binding with DOR [9,10,11,12]. DOR is a seven transmembrane domain receptor coupled with G protein (GCPR) coded by the *ORPD* gene. The agonists of this receptor have demonstrated potent analgesic properties as well as the ability to activate several intracellular pathways, such as AC, different phospholipases, Erk1/2, Akt, c-Jun as well as Ca^2+^ signaling [13]. Recent in vitro studies showed that GEs can cross the intestinal epithelium without cytotoxic effects and increasing cell viability, suggesting a possible role in metabolic and proliferative processes of enterocytes [14,15]. It has been demonstrated that DOR receptor is over-expressed on the plasma membrane during inflammatory processes [15,16,17,18,19] as well as in cancer [20,21,22,23,24,25]. This characteristic may play an important role in the onset of CD comorbidities, specifically enteropathy-associated T-cell lymphoma (EATL) and small bowel carcinoma (SBC), which are associated with undiagnosed CD or not strictly adherent gluten-free diet (GFD) [26,27]. 

The aim of this work was to investigate the molecular and cellular effect of GEs in established SUP-T1 and Caco-2 cells, respectively, as in vitro models of lymphoblastic T-cells and intestinal epithelial barrier, in order to suggest possible functional interplays of GEs in the context of CD and on its associated cancer comorbidities.

## 2. Results

### 2.1. SUP-T1 and Caco-2 Cells Differentially Express DOR 

A preliminary analysis was performed to study the levels of DOR expression in SUP-T1 and Caco-2 cells. DOR protein expression was assessed through immunofluorescence and immunoblotting analyses. As reported in Figure 1, DOR was expressed in the investigated cell lines. Particularly, immunoblotting analysis showed that SUP-T1 expressed significantly higher levels of DOR compared with Caco-2 cells. Immunoblotting analysis was also performed to investigate DOR expression levels on normal human primary fibroblasts, which showed significantly lower levels compared with SUP-T1 cells. 

### 2.2. GEs Treatment Promoted SUP-T1 Cells Proliferation 

Based on the higher DOR expression profile, GEs effects were primary investigated in SUP-T1 cells. To this purpose, cell proliferation was initially evaluated adding GEs at the concentrations reported in BIOPEP, a database in which all the biologically active peptides and their EC_50_ are described. Cell proliferation was investigated by means of cytofluorimetric analysis at 24 h post-treatment (p.t.) after GEs administration to the culture media. As showed in Figure 2A, an increasing trend in SUP-T1 cell proliferation was observed after the addition of GEs compared with non-treated (NT) cells. Particularly, SUP-T1 proliferation was significantly higher following A5, B4, B5 and C5 treatments. Moreover, based on their reduced employed concentrations, 3.4 and 0.02 μM, respectively, B4 and B5 exerted the major effects compared with the other ones (i.e., A4, A5 and C5). The same analysis was also performed on normal human primary fibroblasts to understand whether GEs were able to induce proliferation in normal cells. As reported (Figure 2B), a slight and not statistically significant increase in cell proliferation was scored. 

Considering the observed increase in cell proliferation, cell cycle progression in synchronized SUP-T1 cells was investigated by means of cytofluorimetric analysis after GEs treatments. Specifically, SUP-T1 cells were synchronized using colcemid, a derivative of colchicine capable of depolymerizing microtubules and arresting cells in metaphase [28]. After the removal of the cell cycle block, GEs were added to the culture media for 24 h. Cells in which the cell cycle was restored after colcemid administration (i.e., sample R) were used as the control. As shown in Appendix A, after GEs treatments a statistically significant decrease in the number of cells in G2/M was detected as well as a relative significant increase in G0/G1 phases compared with sample R. Again, based on their reduced employed concentrations, B4 and B5 exerted the major effects compared with the other assayed GEs. These variations in cell cycle activity following GEs administration, according to the concentrations reported in BIOPEP database, were further confirmed by a cytofluorimetric evaluation of Cyclins E2 and A expression in SUP-T1 cells, as documented in Appendix A. In particular, compared to untreated cells, GEs induced a general increase in Cyclins expression. 

Alterations in cell viability/metabolism were then investigated starting from 1 to 4 h p.t. to reveal the cellular kinetic effects of GEs. As reported in Figure 3, an increase in cell viability/metabolism was observed following all GEs treatments although no significant differences were detected. 

Considering the absence of significance in cell metabolic activity as well as the temporal bias that characterized the activity of GCPRs [29], increasing concentrations of these peptides (specifically, 5- to 10-fold compared to those reported in the BIOPEP database) were tested and cell viability was investigated at the same time intervals (i.e., 1 and 4 h p.t.) through MTS assay. As shown in Figure 4A, no cytotoxicity was detected. Conversely, a statistically significant concentration-dependent increase in cell metabolism was observed in SUP-T1 cells treated with exorphins A4 and A5 at all the tested concentrations at 4 h p.t. Since no significance was obtained for B4, B5, and C5 increasing their concentration 5- and 10-fold times, other concentrations were tested using the same experimental scheme. As shown in Figure 4B, MTS assay led to the identification of the effective concentrations for all the three tested GEs (i.e., B4 = 50 µM, B5 = 20 µM, and C5 = 250 µM).

### 2.3. GEs Treatment Induced Mitogenic and Pro-Survival Pathways in SUP-T1 Cells

The minimum concentrations that were able to increase the cell metabolic activity (Figure 4A,B) were then tested though immunoblotting to study the induction of mitogenic and pro-survival pathways, specifically phosphorylation levels of Akt and Erk1/2 kinases. As reported in Figure 5, a significant increase in Akt phosphorylation ratio was observed following GEs administration. Erk1/2 phosphorylation levels were generally increased following GEs treatments, with however statistical significance only in A4, A5 and B4 samples. 

Subsequently, GEs were tested to investigate the activation of pro-survival pathways through the analysis of specific markers, i.e., the anti-apoptotic protein Bcl-2 [30] and the autophagy marker LC3-II [31], after 4 h p.t. As represented in Figure 6, all GEs treatments induced a significant increase in Bcl-2 levels. LC3-II levels showed an increasing trend but significant only for the B5 and C5 treatments. Since GEs were described as agonists of ORs, immunoblotting analysis was performed to investigate alterations in DOR expression levels after GEs treatment but no differences were detected.

### 2.4. B5 Exorphin Induced Mitogenic and Pro-Survival Pathways in Caco-2 Cells

Considering that B5 was described as the most potent gluten exorphin [11], this peptide was tested through MTS assay in Caco-2 cells using the BIOPEP concentration of 0.02 μM. After 4 h p.t., no statistically significant increase was observed in cell metabolic activity. Again, MTS assay was performed at 4 h p.t. using the same concentrations tested in SUP-T1 cells. As represented in Figure 7, a concentration-dependent increase in cell viability was observed in Caco-2 cells, particularly for higher concentrations (>250 μM). 

Similar to SUP-T1 cells, the expression of Akt, Erk1/2, and LC3-II proteins was investigated through immunoblotting in Caco-2 cells following different B5 micromolar administration. Furthermore, STAT3 phosphorylation was studied due to the role of this protein in intestinal epithelial cells (IECs) homeostasis [32]. An increasing trend in Akt phosphorylation was scored although not statistically significant (Appendix A). Conversely, as shown in Figure 8, an increasing trend in Erk1/2 phosphorylation was detected, although statistically significant starting from 250 µM of B5. Moreover, a decrease in LC3-II and STAT3 phosphorylation levels was observed using B5 concentrations at 400 and 500 µM. 

### 2.5. Gluten B Exorphins Did Not Induce Increase in Viability in Normal CD3+ and CD8+ T-Lymphocytes 

The most effective B4 and B5 exorphins were selected and preliminary tested on primary T-cells isolated from 3 healthy subjects to determine their effects on viability at the same time intervals (i.e., 0 to 4 h p.t.) previously investigated in SUP-T1 cells. Specifically, B exorphins were investigated on PBMC-derived bulk CD3+ and on CD8+ cell subsets isolated by labeling with antigen-coated microbeads and magnetic capture. T cells viability was evaluated in a time-kinetics interval by MTS assay, following BIOPEP database concentrations (i.e., 3.4 µM for B4 and 0.02 µM for B5) administration. As reported (Figure 9), in general, B4 and B5 exorphins induced a reduction on T-cells viability. In particular, B4 produced a slight but significant decrease in viability at 50 µM concentration at 30, 60 and 120 min in CD3+ and no significant reduction in CD8+ cells, respectively (Figure 9A,B). In B5 treatment at 20 µM, only CD3+ cells displayed a significant reduction in viability at 30, 60 and 120 min intervals, while CD8+ displayed a significant decrease at 60 and 120 min (Figure 9C,D).

### 2.6. B5 and A4 Showed a Differential DOR Dynamic Profile of Activation in Computational Models

The molecular determinants of the interaction between two selected GEs and DOR were investigated by means of Molecular Dynamics simulations. Multiple 1 µs trajectories were run for B5-DOR and A4-DOR complexes, using Leu-enkephalin-DOR as a control complex, after docking the three peptides into the binding site of the active DOR conformation (PDB code: 6PT2). The equilibrium poses of the three peptides, extracted as representative cluster centers after clustering each dataset according to the ligand position, are shown in Figure 10A. They all share the interaction between the peptide N terminal domain and Asp128, its C terminal domain interacting with Arg291, as well as the peptide Tyr1 pointing to His278. The most significant variability among the complexes is observed at extracellular loop EL3, with Trp284 strongly interacting with B5 Trp4 and less with Leu-enkephalin Phe4. A4 occasionally shows the least stable coordination with the binding site residues and the largest distance from Trp284 (Appendix A). Correspondingly, the structure of the intra-cellular side of the receptor bound to GEs is modulated with respect to control (Figure 10B). B5 is stabilized in a structure more like the Leu-enkephalin complex and the reference agonist bound crystal structure 6PT2, albeit exploring also a wider TM6-TM3 distance (Figure 10B). In contrast, A4 drives the TM6-TM3 distance even further, thereby also occasionally affecting the known GPCR microswitches such as the toggle switch CWxP at Trp274^6.48^ and the NPxxY motif at Tyr318 ^7.53^ [33]. These findings confirm that both GEs can sustain an active state of the receptor, but B5 shows a complex stability more closely resembling Leu-enkephalin, which might be connected to its stronger activity. 

## 3. Discussion

To date, the effect of GEs released from the digestion of gluten was not deeply investigated. However, it has been demonstrated that GEs exert different effects in several physiological context, also depending on their ability to cross the intestinal epithelial barrier [14,15]. Specifically, it has been demonstrated that these peptides can bind ORs with different affinity, as reported by Zioudrou and colleagues [8]. As experimentally reported, exorphins A4 and A5 showed selectivity for the δ-opioid receptor (DOR) but not for κ (KOR); B4 and B5 showed affinity for δ- and μ-opioid receptor (MOR) but not for κ. Among these peptides, B5 was the most potent, with an EC_50_ of 0.05 μM for Guinea pig ileum (GPI) assay and 0.017 μM for MVD assay. On the opposite, B4 was less potent because of the lack of Leucine residue in position 5 [7,11,12].

Since gluten is the trigger of CD and different gluten derived peptides exert immunogenic and cytotoxic effects responsible for CD pathogenesis [1], GEs could be an interesting candidate to investigate in the context of this disorder. Specifically, the opioid activity of GEs could be implicated in the onset of the asymptomatic form of CD. As widely reported in the literature, DOR expression levels increase on the plasma membrane of intestinal cells during inflammatory processes; moreover, this over-expression is also mediated by DOR activation in a positive feedback loop [16,17,18,19]. These mechanisms could contribute to GEs activity in CD thus leading to the inhibition of abdominal pain that characterized the classical CD [6].

Considering these premises as well as that tumor comorbidities are associated with undiagnosed/untreated CD patients [1,26,27], the present work investigated the role of GEs in SUP-T1 and Caco-2 cells, after verifying the expression of DOR by immunofluorescence and immunoblotting. Considering that DOR agonists lead to the activation of proliferative pathways, such as PI3K/Akt and Erk1/2 [13,34,35,36,37], proliferation assays were performed on SUP-T1 cells using the concentration reported in the BIOPEP database. The obtained results demonstrated that GEs stimulate the proliferation of this cell line and, considering the concentration analyzed, confirmed the potent action of B and C compared with A gluten exorphins [11]. Furthermore, as highlighted by the significance levels, with five amino acids (i.e., A5 and B5) exerted a higher effect compared with their counterpart of four amino acids (i.e., A4 and B4). These results were also corroborated by a cell cycle analysis on synchronized SUP-T1 cells. Conversely, no proliferative effects were observed in treated normal human primary fibroblasts although the expression of DOR was verified through immunoblotting. A possible hypothesis for this result could rely on the fact that DOR stimulation in fibroblast leads to the activation of p70(sK6), a protein implicated in cytoskeletal modifications as well as in cell migration, two processes involved in wound healing [38,39]. Subsequently, MTS assays using increasing concentration of GEs were performed starting from 1 to 4 h p.t. to study variation in cell metabolism. This temporal window was selected since mitogenic pathways are activated by ORs agonists in minutes/hours and the time-duration of the molecular response generated by GPCRs stimulation is influenced by different factors [29]. The collected results confirmed the increase in cell metabolism of SUP-T1 cells at 4 h p.t., thus leading to the identification of the minimum effective concentration. Considering the role of PI3K/Akt and Erk1/2 in cell proliferation as well as in DOR signaling pathway [13], phosphorylation of these two kinases was investigated by immunoblotting after 4 h p.t., accordingly to the MTS results. As showed, phosphorylation of Akt and Erk1/2 was observed in all GEs treatments. Furthermore, the levels of Bcl-2 were investigated to determine the effect of GEs treatments on apoptosis. Recent studies showed that IL-15 triggers anti-apoptotic pathways in intraepithelial lymphocytes (IELs) derived from RCDII patients through Bcl-2 and/or Bcl-XL [40]. As showed, all GEs led to an increase in Bcl-2 expression levels. Due to the crosstalk between autophagy and apoptosis [41] as well as in relation to the autophagy impairment caused by digested gliadin [42,43], the autophagy marker LC3-II was investigated. As reported, GEs treatments led to an increase in LC3-II levels of SUP-T1 cells, suggesting an increase in autophagy process induced by Erk1/2 activation [44] as well as by AMPK [45] due to the decreased levels of ATP generated by AC, a protein down-stream of DOR [6].

Considering the collected results as well as the evidence present in literature, B5 exorphin was selected and investigated in Caco-2 cells. Increasing concentrations of this peptide led to an increase in cell metabolism and to Erk1/2 activation. Moreover, a reduction in LC3-II levels was observed at the concentrations of 400 and 500 µM. This result could suggest a high autophagy flux and, as consequent, a massive degradation of LC3-II as well as an early blockage in the autophagy pathway. This second hypothesis is also suggested by the evidence collected in our laboratory that digested gliadin causes autophagy blockage in Caco-2 cells [42,43]. Lastly, STAT3 phosphorylation levels were assayed after B5 treatment due to its important role in IECs homeostasis. Particularly, STAT3 activation is implicated in wound healing, pathogen defense, and intestinal barrier integrity [46,47,48,49]. As reported, B5 treatment led to a reduction in STAT3 activation, thus leading to the hypothesis that GEs could contribute to the detrimental effects exerted by gliadin on IECs.

The results of the effects of GEs in reducing viability in human naïve CD3+ and CD8+ T cells agreed with those of DOR agonists that inhibited proliferation of murine T-cells [50]. Furthermore, Leu- and Met-enkephalins, that share sequence similarities with GEs, induced different effects on immune cells, according to their concentrations or to T-cell mitogenic status [51,52,53].

According to the data in the literature regarding the effects of enkephalins in immune cells, the behavior of GEs could be influenced by the metabolic status of the cells, as in the case of activated T-cells in inflammation [54,55], as well as by the DOR receptor levels on the plasma membrane, whose levels in T-cells are affected by different activating status [56,57,58].

Altogether, these results suggested a possible role of GEs in CD pathogenesis. Particularly, GEs could contribute to the induction of lymphocytes activation and infiltration as well as promoting IECs loss of homeostasis. Both these conditions could contribute to the maintenance of the CD inflammatory status and, possibly, to the onset of the asymptomatic form of CD. Future experiments might be aimed at evaluating GEs effects on human primary lymphocytes derived from CD patients as well as in organ cultures and to further dissect the molecular effects of GEs.

## 4. Materials and Methods

### 4.1. Cell Cultures and GEs Administration

SUP-T1 and Caco-2 cells were purchased from the American Type Culture Collection (ATCC). SUP-T1 were cultured in RPMI-1640 medium (Euroclone, Milano, Italy) supplemented with 10% FBS, 100 U/mL penicillin, 0.1 mg/mL streptomycin (Sigma, St. Louis, MO, USA), 1% Non-Essential Aminoacids (Euroclone) at 37 °C in a 5% CO_2_ atmosphere. Caco-2 cells and normal human primary fibroblasts, isolated as reported [59], were cultured using DMEM medium (Euroclone). SUP-T1 and human primary fibroblasts were seeded at the final concentration of 1 × 10^5^/mL whereas Caco-2 at the final concentration of 2 × 10^5^/mL. Lyophilized GEs were purchased from ThermoFisher (Rockford, Illinois, USA), resuspended in anhydrous DMSO (BDH Chemicals, Radnor, PA, USA) and added to cell cultures 24 h after their seeding.

Peripheral blood mononuclear cells (PBMCs) were obtained from healthy donors after informed consent (Cell Factory, Policlinico S.Matteo, Pavia, Italy) by Ficoll-Lympholite (Cedarlane, Burlington, Ontario, Canada) density gradient separation. PBMCs were used in bulk (CD3+), or further separated into CD8+ cell subsets by labeling with antigen-coated microbeads and magnetic capture (Miltenyi Biotec, Bergisch Gladbach, Germany). After performing purity control by flow cytometry, cells were seeded in flat-bottomed 96 well plates at a concentration of 3 × 10^4^ cells/mL in RPMI-1640 medium (Euroclone) supplemented with 10% FBS, 100 U/mL penicillin, 0.1 mg/mL streptomycin (Sigma) for 24 h before GEs administration.

GEs sequences, molecular weight and purity grades are reported in Appendix A. GEs stocks were kept at −80 °C. GEs treatments were initially performed according to the EC_50_ values reported in the BIOPEP database, in which are reported all the bioactive peptides and the relative EC_50_ [60]. Subsequently, other GEs concentrations were assayed through MTS assay.

### 4.2. Immunofluorescence Analysis

Immunofluorescence were performed as reported [61]. Specifically, SUP-T1 cells (2.5 × 10^5^) were seeded in multiwell-6 plate in a final volume of 2.5 mL of RPMI. After 24 h they were collected, centrifuged at 300× *g* for 5 min and washed with PBS. Then, cells were fixed with ice-cold fixative solution (methanol and acetic acid, 3:1) for 20 min at −20 °C, added on a coverslip and dried at room temperature (RT). After this step, cells were incubated one hour at RT with a monoclonal anti-DOR primary antibody (Abcam, Cambridge, UK) diluted 1:60 in 5% non-fat milk (*w/v*) in T-TBS (138 mM NaCl, 20 mM Tris-HCl pH 7.6, 0.1% Tween-20). Subsequently, cells were washed three times with T-TBS and incubated with a species-specific polyclonal secondary antibody conjugated with AlexaFluor-488 at a 1:60 final dilution in 5% non-fat milk in T-TBS. Finally, the coverslip was washed three times with T-TBS and treated with ProLong Gold antifade reagent with DAPI (Invitrogen, Carlsbad, CA, USA) according to manufacturer’s instructions. Caco-2 cells were treated in the same way using primary antibodies against DOR and α-tubulin. Fluorescent signal was observed using LEICA DM6 B fluorescent microscope with ORCA-Flash 4.0 V3 system (Leica, Wetzlar, Germania).

### 4.3. MTS Viability Assay

Cells viability was tested using the CellTiter 96 AQueous One Solution Cell Proliferation Assay kit (Promega, Madison, WI, USA) according to manufacturer’s instructions. Cells (1 × 10^4^) were seeded in multiwell-96 plate in 100 µL of growth medium. GEs were added after 24 h, and cells viability was tested after different time intervals by adding 20 µL of MTS reagent. Absorbance (490 nm) was detected using the microplate reader TECAN Sunrise (Männedorf, Switzerland). Each treatment was analyzed using at least 8 wells for each treatment.

### 4.4. Cytofluorimetric Analysis

Cells proliferation was assayed using the Muse Count & Viability Assay (Luminex), according to manufacturer’s instructions. Cells (1 × 10^5^) were seeded in a multiwell-24 plate in 1 mL of complete medium. GEs were added and cells collected 24 h p.t., washed with PBS and resuspended in 50 µL of PBS and 450 µL of Muse Count & Viability Reagent. After an incubation of 5 min at RT, cells were analyzed at the cytofluorometer.

Cell cycle analysis was performed as described [62]. Cells (2.5 × 10^5^) were seeded in a multiwell-6 plate in 2.5 mL of complete medium. Colcemid (10 µg/mL) was added to synchronize cells in metaphase. After 24 h, colcemid was removed by substituting the culture medium and GEs were administered for additional 24 h. Cells were then collected, centrifuged at 300× *g* for 5 min and fixated overnight with 200 µL of 70% ethanol. Subsequently, cells were centrifuged and resuspended in propidium iodide (50 µg/mL) and RNAse A (0.1 mg/mL). The samples were analyzed after an overnight incubation at 4 °C by BD FACSLyrics (BD Bioscience, Franklin Lakes, NJ, USA).

Rabbit polyclonal Cyclin E2 (clone H-140) antibody form Santa Cruz Biotechnologies (Dallas, TX, USA) and a mouse monoclonal Cyclin A (clone CY-A1) antibody (Sigma) were used to analyze cell cycles as follows: 2 μg of each antibody were fluorescently conjugated using DyLight555 labelling kit (Biorad, Hercules, CA, USA) as described [63]. SUP-T1 cells (about 10^5^ for each sample), treated for 24 h with the following micromolar amounts of GEs (i.e., A4 = 70; A5 = 60; B4 = 3.4; B5 = 0.02; C5 = 13.5, according to BIOPEP database) were then fixed and permeabilized using ice cold methanol for 15 min, washed twice with ice cold PBS and then blocked with PBS containing 1% Fetal Calf Serum (FCS) for 30 min at room temperature. After PBS washing, cells were incubated with 2 μL of each fluorescently labelled antibody in a PBS + FCS 1% volume of 50 μL for 2 h at room temperature. Finally, cells were washed twice with D-PBS and analyzed for fluorescence intensity using Guava Muse Cell Analyser (Luminex, Austin, TX, USA), analyzing 2000 events in two independent replicas.

### 4.5. Immunoblotting Analysis

Cells (2.5 × 10^5^) were cultured in a multiwell-6 plate with 2.5 mL of complete medium. GEs were added and cells collected after 4 h p.t. Immunoblotting was performed as described [64]. Cells were collected and lysated in ice-cold Triton X-100 (50 mM Tris-HCl pH 7.4, 150 mM NaCl, 1% Triton X-100) supplemented with Complete Mini Protease Inhibitor cocktail 7X (Roche, Basel, Switzerland) and sodium orthovanadate 1 mM (Sigma). Proteins were quantified using the Quant-It Protein Assay Kit (Invitrogen, Carlsbad, CA, USA). Proteins (20–30 µg) were added to Laemmli sample buffer (2% SDS, 6% glycerol, 150 mM β-mercaptoethanol, 0.02% bromophenol blue and 0.5 M Tris-HCl pH 6.8), denaturated for 5 min at 95 °C and separated on 12% SDS-PAGE according to protein size. After electrophoresis, proteins were transferred onto nitrocellulose membrane using the Trans-Blot Turbo Transfer System (Biorad) according to the manufacturer’s instructions. The membranes were then blocked 1 h at room temperature (RT) with 5% (*w/v*) non-fat milk in T-TBS (138 mM NaCl, 20 mM Tris-HCl pH 7.6, 0.1% Tween-20) and incubated overnight at 4 °C with monoclonal primary antibodies against DOR (Abcam, ab176324), Erk1/2 (#9102), phosphor-Erk1/2 (Tyr204/Tyr187, #5726), Akt (#9272), phopsho-Akt (Ser473, #9271), Bcl-2 (#9941), phospho-Bcl-2 (Ser70, #9941), STAT3 (#9139), phosphor-STAT3 (Tyr705, #9145), and LC3-II (#2775) (Cell Signaling, Danvers, MA, USA) diluted at 1:2000 in 5% non-fat milk in T-TBS, whereas monoclonal primary antibody against α-tubulin (Abcam, ab7291) or β-actin (Cell Signaling, #5125) were diluted at 1:4000. Species-specific peroxidase-labelled ECL secondary antibodies (Cell Signaling, 1:2000 dilution) were used in 5% non-fat milk in T-TBS. Proteins signals were detected using the ECL Prime Western Blotting Detection Kit (GE Healthcare) by means of Chemidoc MP (Biorad). Densitometric analysis was conducted with ImageJ software (http://rsbweb.nih.gov/ij (accessed on 1 October 2022)).

### 4.6. Molecular Dynamics Simulations

The peptide–receptor complexes were modelled starting from the active DOR structure bound to the agonist peptide KGCHM07 (PDB code: 6PT2) [65]. The standard Peptide docking protocol of Schrodinger [66] was applied, using default settings and MM-GBSA scoring, centering the grid on the crystal peptide ligand. Poses were analyzed and selected among the most favorable docking scores. The starting complexes of Leu-enkephalin (YGGFL), B5 (YGGWL), and A4 (GYYP) were chosen with Y and F/W/Y orientation in common. Each complex was embedded in a 90:10 POPC CHOL membrane and solvated in a cubic box containing K + Cl- corresponding to 150 nM ionic concentration and a total of 103,000 atoms. Using NAMD version 2.10 [67] and CHARMM36m force field, the three systems were first subjected to the multi-stage equilibration protocol as in [68]. Production runs were performed with OpenMM 7.4 software [69] using PME for electrostatic interactions, at 310 K temperature, under NPT ensemble using semi-isotropic pressure coupling, and with 4fs integration time-step (with mass repartitioning). Monte Carlo barostat and Langevin thermostat were used to maintain constant pressure and temperature, respectively. The van der Waals interactions were calculated applying a cutoff distance of 12 Å and switching the potential from 10 Å. Six replicates of 1 µs each were generated for each system, leading to a total of 18 µs simulation time. Clustering analysis was carried out using the GROMACS clustering tool and the Gromos method [70] with cutoff 0.2 nm.

### 4.7. Statistical Analysis

The data were analyzed using the statistics functions of the MedCalc statistical software version 18.11.6. (http://www.medcalc.org (accessed on 3 November 2022)). The ANOVA-One Way test differences were considered statistically significant when *p* < 0.05.

## Figures and Tables

**Figure 1 ijms-24-03912-f001:**
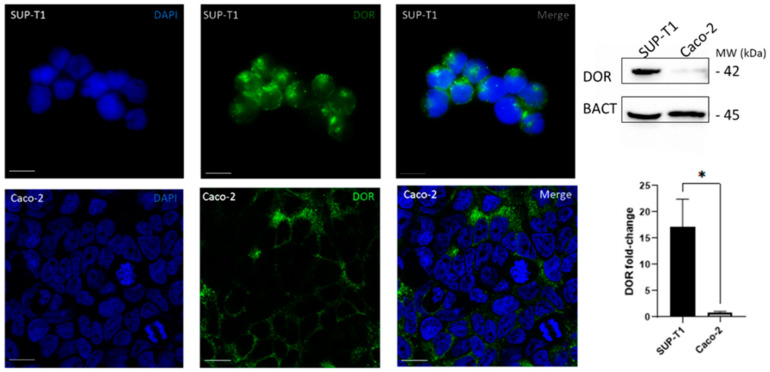
Analysis of DOR expression levels. On the left, immunofluorescence of DOR (green) in SUP-T1 and Caco-2 cell lines. Nuclei were stained in DAPI. Scale bars (10 µm) are reported. Images were visualized through a LEICA DM6 B fluorescent microscope with ORCA-Flash 4.0 V3 system (60X). On the right, immunoblotting and relative densitometric analysis of DOR in SUP-T1 and Caco-2 cells. DOR expression was normalized to β-actin (BACT) levels. Normalized values are shown on the *Y*-axis as arbitrary units. Molecular weights (MW) and SEM bars are reported. Asterisk indicates *p* < 0.05 (ANOVA One-Way), compared with SUP-T1 cells. The experiments were performed on three independent biological replicas.

**Figure 2 ijms-24-03912-f002:**
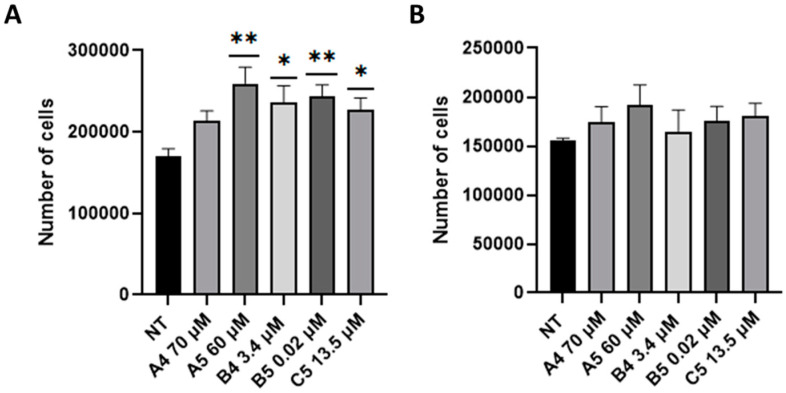
Analysis of cell proliferation following GEs administration. (**A**) SUP-T1 and (**B**) normal human primary fibroblasts cell proliferation after GEs administration at 24 h p.t. GEs were used according to the concentration reported in the BIOPEP database. Cytofluorimetric analysis was performed through Muse Cell Analyzer. Asterisks indicated *p* < 0.05 (*) or *p* < 0.01 (**), compared with non-treated (NT) cells, ANOVA One-Way test. Analysis was performed on four independent biological replicas with 1000 events collected each. SEM bars are reported.

**Figure 3 ijms-24-03912-f003:**
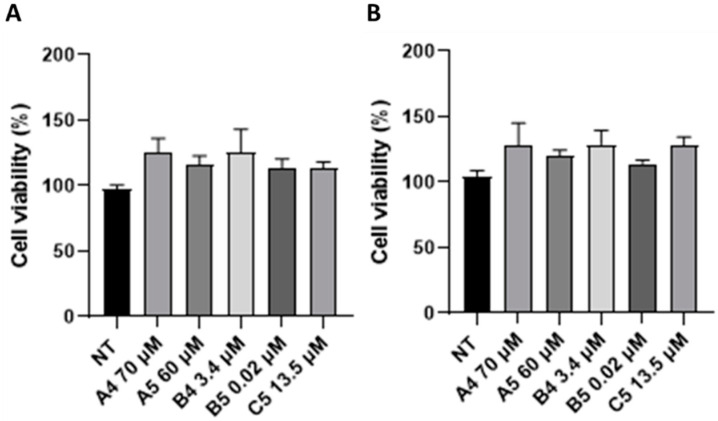
MTS analysis of SUP-T1 cell viability/metabolism at (**A**) 1 and (**B**) 4 h p.t. GEs were used according to the concentration reported in the BIOPEP database. Analysis was performed on three independent biological replicas. SEM bars are reported.

**Figure 4 ijms-24-03912-f004:**
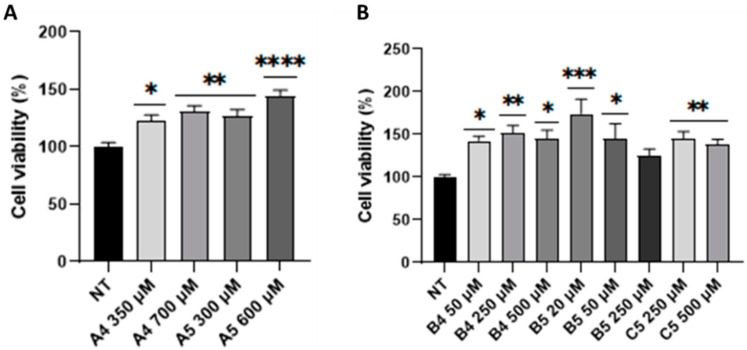
MTS assay on SUP-T1 cells at 4 h after single GEs treatments (**A**,**B**). Asterisks indicated *p* < 0.05 (*), *p* < 0.01 (**), *p* < 0.001 (***), and *p* < 0.0001 (****) compared with non-treated (NT) cells, ANOVA One-Way test. Analysis was performed on three independent biological replicas. Data are reported as mean ± SEM.

**Figure 5 ijms-24-03912-f005:**
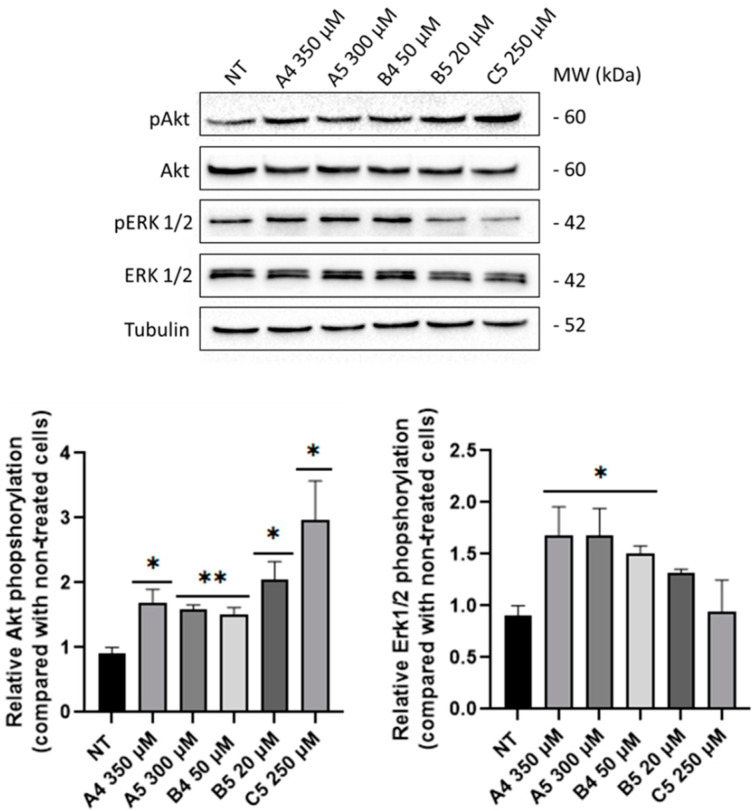
Akt and Erk1/2 phosphorylation levels in SUP-T1 cells at 4 h p.t. after GEs administration at the following concentrations: A4 = 350 μM, A5 = 300 μM, B4 = 50 μM, B5 = 20 μM and C5 = 250 μM. Akt and Erk1/2 were analyzed through immunoblotting and relative densitometric analyses (lower panels). Phosphorylated proteins were normalized to the total proteins and housekeeping levels. Normalized values are reported on *Y*-axis as arbitrary units. Molecular weights (MW) in kDa and SEM bars are shown. Asterisks indicate *p* < 0.05 (*) or *p* < 0.01 (**), compared with non-treated (NT) control cells (ANOVA One-Way). The experiments were performed on four independent biological replicas.

**Figure 6 ijms-24-03912-f006:**
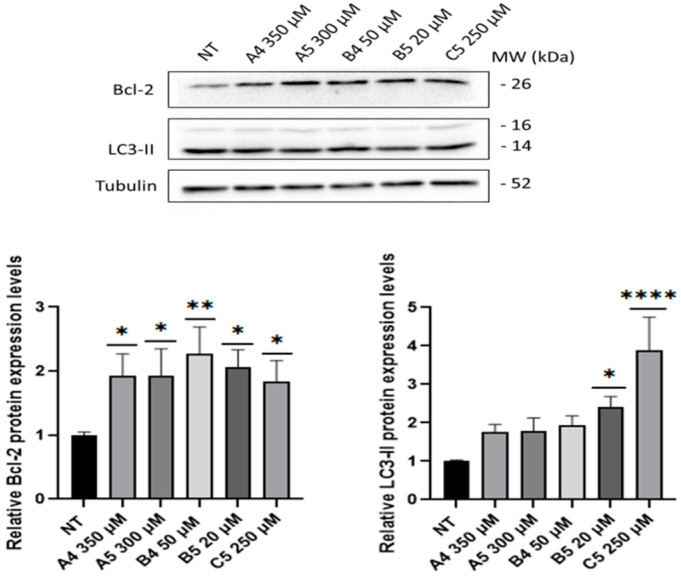
Bcl-2 and LC3-II levels in SUP-T1 cells at 4 h p.t. after GEs treatment: A4 = 350 μM, A5 = 300 μM, B4 = 50 μM, B5 = 20 μM and C5 = 250 μM. All the proteins were analyzed through immunoblotting and relative densitometric analyses. Bcl-2 and LC3-II were normalized to housekeeping levels. Bcl-2 proteins were normalized to housekeeping levels. Normalized values are reported on *Y*-axis as arbitrary units. Molecular weights (MW) in kDa and SEM bars are shown. Asterisks indicated *p* < 0.05 (*), *p* < 0.01 (**), *p* < 0.0001 (****), compared with non-treated (NT) control cells (ANOVA One-Way). Bcl-2 and LC3-II levels were tested on four independent biological replicas.

**Figure 7 ijms-24-03912-f007:**
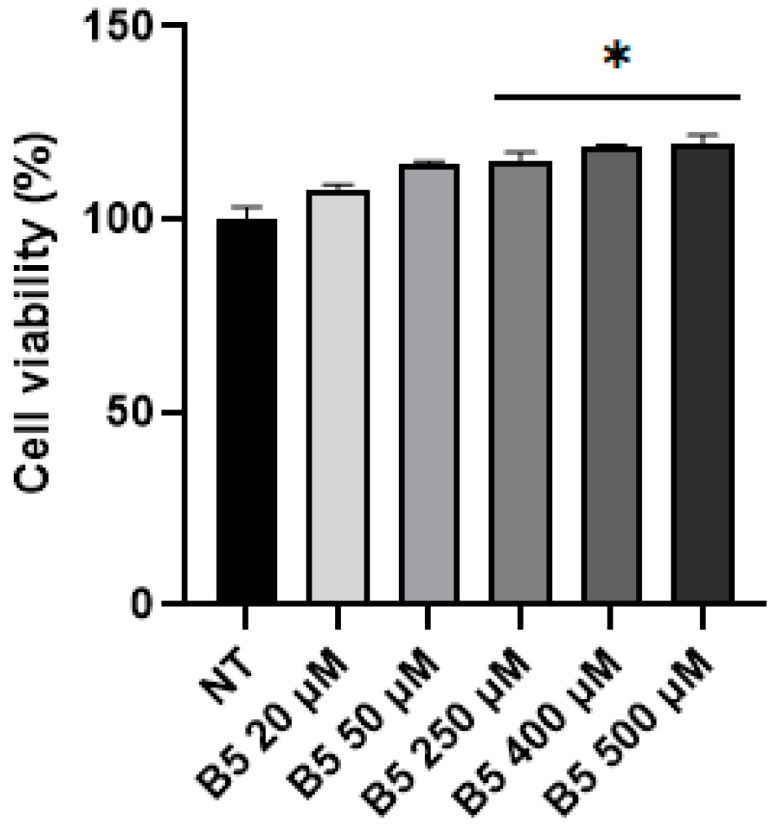
MTS assay on Caco-2 cells at 4 h p.t. Asterisk indicated *p* < 0.05 (*) compared with non-treated (NT) cells, ANOVA One-Way test. Analysis was performed on three independent biological replicas. Data are reported as mean ± SEM.

**Figure 8 ijms-24-03912-f008:**
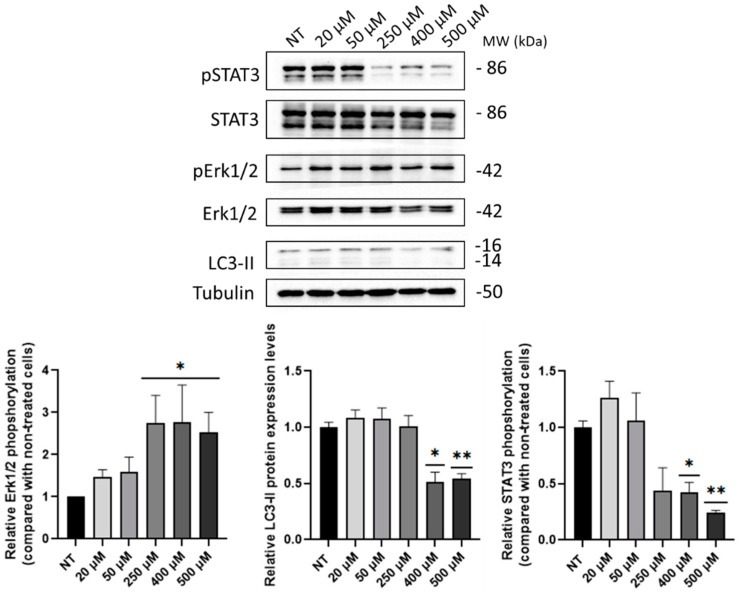
Phospho-Erk1/2, LC3-II, and phospho-STAT3 levels in Caco-2 cells at 4 h p.t. after B5 administration at increasing concentrations (i.e., 20, 50, 250, 400 and 500 μM). Phosphorylated proteins were normalized to the total proteins and housekeeping levels. Normalized values are reported on *Y*-axis as arbitrary units. Molecular weights (MW) in kDa and SEM bars are shown. Asterisks indicated *p* < 0.05 (*) or *p* < 0.01 (**), compared with non-treated (NT) control cells (ANOVA One-Way). The experiments were performed on three independent biological replicas.

**Figure 9 ijms-24-03912-f009:**
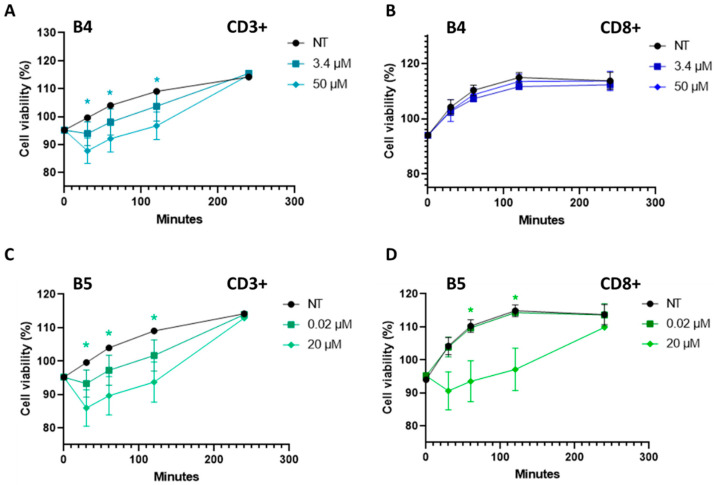
MTS time-kinetics viability assays of human normal primary CD3+ and CD8+ T-cells after B4 and B5 administration. (**A**) CD3+ T-cells viability after B4 treatment (3.4 and 50 µM). Cyan asterisks indicate *p* < 0.05, ANOVA One-Way, in 50 µM treated samples compared with non-treated (NT) cells at 30, 60 and 120 min. (**B**) CD8+ T-cells viability after B4 treatment (3.4 and 50 µM). (**C**) CD3+ T-cells viability after B5 treatment (0.02 and 20 µM). Green asterisks indicated *p* < 0.05, ANOVA One-Way, in B5 20 µM compared with non-treated (NT) cells at 30, 60 and 120 min. (**D**) CD8+ T-cells viability after B5 treatment (0.02 and 20 µM). Green asterisks indicated *p* < 0.05, ANOVA One-Way, in B5 20 µM compared with non-treated (NT) cells at 60 and 120 min. All the experiments were performed on three independent biological replicates, using human naïve T-cells isolated from 3 normal subjects. Data are reported as mean ± SEM.

**Figure 10 ijms-24-03912-f010:**
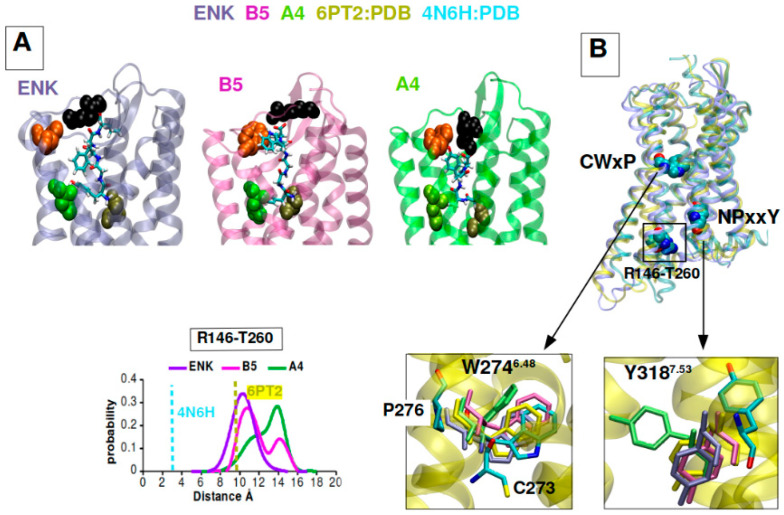
Computational models of GEs-DOR interactions. (**A**) Representative complexes of Leu-Enk-DOR, B5-DOR and A4-DOR extracted from simulations, showing the main contact points between peptide ligands and the receptor. (**B**) Superposition of agonist-bound crystal structure (6PT2), natrindole-bound crystal structure (4N6H) and representative structure of the Leu-enkephalin simulation, highlighting the micro-switches CWxP and NPxxY and residues 146 and 260 on TM3 and TM6 respectively. Bottom, left: distance histogram showing the distribution of distances between Cα atoms of R146 and T260. Right: Close-up superposition of W274 (toggle switch) and Y318 in the inactive natrindole structure (cyan) and in simulation representative snapshots.

## Data Availability

Not applicable.

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
