# Peer review of "Gluten Exorphins Promote Cell Proliferation through the Activation of Mitogenic and Pro-Survival Pathways"

_ijms, 2023, doi:10.3390/ijms24043912_

Round 1

Reviewer 1 Report (Previous Reviewer 1)

Manai et al. investigated the cellular and molecular effects of gluten exorphins (GEs) to understand the functional role of these in Celiac disease. GEs are small peptides derived from digestion of gluten. They tested the effect of five of the GEs in SUP-T1, Caco2 and human primary fibroblast cell lines. Since the GEs bind with delta-opioid receptors (DOR), the authors looked at the expression of these receptors on SUP-T1 and Caco-2 cell lines at the beginning of the study. Then they looked at the proliferation, mitogenic and pro-survival pathways in these cell lines. They also performed computational modeling to study the interaction of A4 and B5 GEs with DOR. This study attempted to explain the impact of GEs in CD pathogenesis using in vitro studies and computational modeling.

The following are some minor comments to improve this manuscript:

Please rephrase the lines (343-344) to improve the readability.

In L50, “the hypothesis” is not fitting in the statement.

In L261 and 271, change “contest” to “context”.

In L301, “as well as” was written as “as we as”.

In L346, change “by“ to “from”.

Author Response

Comments and Suggestions for Authors

Manai et al. investigated the cellular and molecular effects of gluten exorphins (GEs) to understand the functional role of these in Celiac disease. GEs are small peptides derived from digestion of gluten. They tested the effect of five of the GEs in SUP-T1, Caco2 and human primary fibroblast cell lines. Since the GEs bind with delta-opioid receptors (DOR), the authors looked at the expression of these receptors on SUP-T1 and Caco-2 cell lines at the beginning of the study. Then they looked at the proliferation, mitogenic and pro-survival pathways in these cell lines. They also performed computational modeling to study the interaction of A4 and B5 GEs with DOR. This study attempted to explain the impact of GEs in CD pathogenesis using in vitro studies and computational modeling.

The following are some minor comments to improve this manuscript:

Please rephrase the lines (343-344) to improve the readability.

In L50, “the hypothesis” is not fitting in the statement.

In L261 and 271, change “contest” to “context”.

In L301, “as well as” was written as “as we as”.

In L346, change “by“ to “from”.

Thank you for your positive and encouraging review of our manuscript.

We have provided in the revised manuscript all your suggestions.

Reviewer 2 Report (New Reviewer)

interesting pathway

Author Response

Comments and Suggestions for Authors

interesting pathway

Thank you for Your positive and encouraging review of our manuscript.

Reviewer 3 Report (New Reviewer)

In the manuscript (ID: ijms-2173573), Federico Manai et al. revealed that gluten exorphins cell proliferation via mitogenic and pro-survival signaling.

Overall, the manuscript is well-written. However, there are a few suggestions and comments. This will help readers understand better. I recommend considering major and minor revision for publication.

Major points

1.     The authors show gluten exorphins promote SUP-T1 cell proliferation.

The authors use lymphoblastic leukemia cell lines. The authors should examine whether gluten exorphins promotes the normal T cell proliferation. 

2.     The authors need to determine if gluten exorphins is promoting the cell proliferation through d-opioid receptor signaling. For this, the knockdown of d-opioid receptor and the treament with d-opioid receptor inhibitor are necessary.

3.     To confirm gluten exorphins promote the cell proliferation, the authors should examine whether the intravenous administration of gluten exorphins enhances the cell proliferation.  

4.     I would be better if they compared the amount of cyclin after gluten exorphins treatment.

Minor points.

1.     Please confirm figure legend in Figure 6.

p<0.001(****)p<0.0001(****) ?

2.     Page 2, line 68, [14-15] [14,15]

3.     Page 2, line 73, [26-27] [26,27]

Author Response

We have tried to take into consideration the critical issues highlighted. We hope that the Reviewer can first understand that the proposed manuscript is essentially an in vitro study and therefore can constitute a starting point for subsequent and more informative analyses. In vivo experimentation (on patients?) suggested by intravenous administration cannot represent a valid experimental approach as explained below.

We hope that the Reviewer will understand the fact that we have tried to respond as far as possible - experimentally and conceptually - to the critical issues raised. In particular, as analytically reported below and in the new version of the manuscript, we evaluated the effects of gluten exorphins in lymphocyte populations from 3 healthy subjects, demonstrating the absence of induced proliferative effects; we also assayed the expression of two main cyclins confirming the data previously shown in MTS assays, immunoblotting and flow cytometry of GEs induction of proliferation.

In conclusion, appealing to the clemency and understanding of the Reviewer, we believe, as far as possible, to have improved the quality of the manuscript.

Finally, we just wanted to inform the Referee that 4 other fellow Reviewers (two in this phase and two recruited in a previous round for which we received a “Minor Revision“ evaluation) had only requested minor revisions and had reported excellent reviews on the elaborate.

Major points

  1. The authors show gluten exorphins promote SUP-T1 cell proliferation.

The authors use lymphoblastic leukemia cell lines. The authors should examine whether gluten exorphins promotes the normal T cell proliferation.

According to the request, we have provided human naïve lymphocytes (CD3+ and CD8+ subsets) from 3 healthy donors and assayed independently by MTS analysis in different replicas at the time intervals used for the reported established cell line. These additional results were included as new Figure 9 in the revised version of the manuscript, reporting that the assayed gluten exorphins did not induced an increase in T-cells viability. In the discussion section we have commented the obtained results taking into account some evidence collected in literature. 

  1. The authors need to determine if gluten exorphins is promoting the cell proliferation through d-opioid receptor signaling. For this, the knockdown of d-opioid receptor and the treament with d-opioid receptor inhibitor are necessary.

We certainly thank the reviewer for this discussion point. Originally, we also thought about the possibility of DOR gene silencing. However, a series of scientific evidences already published

-Deletion of delta-opioid receptor in mice alters skin differentiation and delays wound healing. Differentiation 2006 Apr;74(4):174-85. doi: 10.1111/j.1432-0436.2006.00065.x.

-Down-regulation of kappa opioid receptor promotes ESCC proliferation, invasion and metastasis via the PDK1-AKT signaling pathway. Cell Communication and Signaling volume 20, Article number: 35 (2022).

-Crosstalk between Delta Opioid Receptor and Nerve Growth Factor Signaling Modulates Neuroprotection and Differentiation in Rodent Cell Models. Int. J. Mol. Sci. 2013, 14(10), 21114-21139;

-Molecular Mechanisms of Opioid Receptor-dependent Signaling and Behavior. Anesthesiology December 2011, Vol. 115, 1363–1381. https://doi.org/10.3390/ijms141021114

have shown how - on different cell lines and in vivo experiments - the knock down or the use of specific inhibitors of the delta opioid receptor can induce significant changes in the cellular pathways such as cell viability, proliferation and differentiation, that are all pathways we investigated as possible effects of gluten exorphins administration.

Similarly, the use of Deltorphin A, a potent and selective agonist for the delta-opioid receptor resulted in CD8+ T cells from C57BL/6 mice inducing a dose-dependently inhibition of proliferation (Antiproliferative effects of delta opioids on highly purified CD4+ and CD8+ murine T cells. Journal of Pharmacology and Experimental Therapeutics June 1995, 273 (3) 1105-1113).

In addition, since the aim of the study was to evaluate the effects of gluten exorphins in cellular contexts as close as possible to the physio-pathological ones, we considered unnecessary to alter (by silencing or overexpressing) the expression of DOR gene to not interfere or mask differences in viability or cells proliferation indexes.

Furthermore, to increase the biological complexity, it is well known that delta opioid receptor-knockout mice showed inflammatory pain phenotypes (Inflammatory pain is enhanced in delta opioid receptor-knockout mice, European Journal of Neuroscience 2008, 27(10):2558-67): as a consequence, the lymphoblastic T-cells we employed, eventually subjected to DOR silencing, might exhibit inflammasome differences that might affect their viability and proliferation rates, confounding the effects induced by gluten exorphins treatments.

  1. To confirm gluten exorphins promote the cell proliferation, the authors should examine whether the intravenous administration of gluten exorphins enhances the cell proliferation.

This suggestion is not feasible for at least two reasons:

The first of a biochemical and physiological nature as gluten exorphins, as known, derive from an enzymatic digestion process in the gastrointestinal tract, following gluten-based food uptake. An intravenous administration of these peptides (also, at what dosage in consideration of the body mass?) would therefore be completely outside the physiological context. Going into the possible detail of an in vivo experiment, as known, celiac disease does not physiologically exist in rodents and, consequently, the mouse species does not constitute a valid experimental model for gluten exorphins administration. The only approach would be the use of human subjects, for which, as already explained, the intravenous administration of these peptides is completely extraneous to the experimental model.

The second reason would obviously be linked to an in vivo design which would require approval by the local Ethics Committee. Going back to the first point, i.e., a method of administration outside the physiological context, the systemic experimentation would certainly not be approved as such, probably resulting in unpredictable pathological off targets effects on subjects also due to the proportional high dosage of the peptides in relation to the body mass. Furthermore, it should also be considered that the possible times for an in vivo trial request, if approved, and the recruitment of subjects would certainly require a long interval of time.

Finally, we have also stress on the fact that this contribution is intended ONLY as an in vitro preliminary study, that, for the first time underlined the activation of proliferative pathways in the investigated cell lines by gluten exorphins.

  1. I would be better if they compared the amount of cyclin after gluten exorphins treatment.

In thanking the Referee for this important suggestion, we evaluated the expression of two of the main cyclins, i.e., A and E2, in SUP-T1 cells treated with gluten exorphins at the concentrations indicated in the BIOPEP database. The results obtained by cytofluorimetric analysis, reported in the new Figure S2, confirm those highlighted by cytofluorimetric analysis of the cell cycle.

All Minor points have been corrected as requested.

Round 2

Reviewer 3 Report (New Reviewer)

Federico Manai and coworkers have made substantial efforts and included new data to answer questions and comments made by the reviewer.

The authors have responded appropriately to my concerns.

Author Response

Dear Editor

I have further verified and introcuded alla the minor revision requested in the uploaded file 

Thanking for this further editing check of the manuscript

Sincerely

Sergio Comincini

This manuscript is a resubmission of an earlier submission. The following is a list of the peer review reports and author responses from that submission.

Round 1

Reviewer 1 Report

IJMS-2 comments

In this manuscript, the authors investigated the cellular and molecular effects of gluten exorphins (GEs). They tested the effect of five of the GEs in SUP-T1 and Caco2 cell lines. Since the GEs bind with delta-opioid receptors (DOR), the authors looked at the expression of these receptors on the test cell lines at the beginning of the study. Then they looked at the proliferation, mitogenic and pro-survival pathways in these cell lines. They also performed computational modeling to study the interaction of A4 and B5 GEs with DOR.

The following are the comments for this manuscript:

 For testing the proliferation (fig2) and viability (fig3) analysis, the authors used the concentration provided in the BIOPEP database. The proliferation analysis was significant for most of them. Then the authors increased the concentration from 5-fold to 1000 folds. These concentrations appear to be very high that may not be bioavailable. How do the authors justify the findings arrived by these high concentrations?

The authors labelled the western blot image quantification values as fold-change. These are not fold-change values but relative intensity values with arbitrary units. Please change the y-axis labels appropriately.

How did the authors fix the SUP-T1 cells on the slides for immunofluorescence analysis as these are non-adherent? The provided experimental detail is not convincing.

STAT3 phosphorylation site was not mentioned in the method section. Please provide the detail. The authors could have provided the catalog number for each of the antibodies.

Expansion of the abbreviation GPI assay was not provided (L247).

The supplementary table 1 was mentioned in the manuscript but not provided.

The authors have to modify the y-axis labels for supplementary figure 1A and 1B to identify the cell cycle phase they refer to. The cell-cycle analysis presented in this has “recovery” data group. Does this is an untreated group?

Author Response

REVIEWER 1 ANSWERS

In this manuscript, the authors investigated the cellular and molecular effects of gluten exorphins (GEs). They tested the effect of five of the GEs in SUP-T1 and Caco2 cell lines. Since the GEs bind with delta-opioid receptors (DOR), the authors looked at the expression of these receptors on the test cell lines at the beginning of the study. Then they looked at the proliferation, mitogenic and pro-survival pathways in these cell lines. They also performed computational modeling to study the interaction of A4 and B5 GEs with DOR.

The following are the comments for this manuscript:

For testing the proliferation (fig2) and viability (fig3) analysis, the authors used the concentration provided in the BIOPEP database. The proliferation analysis was significant for most of them. Then the authors increased the concentration from 5-fold to 1000 folds. These concentrations appear to be very high that may not be bioavailable. How do the authors justify the findings arrived by these high concentrations?

A: We thanks the Reviewer for his/her comments. As described in the literature, the kinetics of the activation of GPCRs is generally affected by temporal/spatial bias. We decided to increase the concentration of GEs from 5- to 1000-folds to stress the activation of downstream pathways, whose activation probably occurred in a shorter temporal window, and to determine the presence of potential cytotoxic effects. The investigated kinases, i.e., Akt and Erk1/2, are involved upstream in the relative pathways and their activation is generally transient. The analysis with high concentration of GEs allowed us to study the phosphorylation of this kinases due to their prolonged activation. Furthermore, in the papers of Fukudome and colleagues (i.e. FEBS Lett. 1992 Jan 13;296(1):107-11. doi: 10.1016/0014-5793(92)80414-c; FEBS Lett. 1997 Aug 4;412(3):475-9. doi: 10.1016/s0014-5793(97)00829-6), it has been demonstrated that some GEs (e.g., A4 and A5) can stimulate the μ-opioid receptors at concentration higher than 1 mM. Overall, these results suggested the mechanism of action of GEs and demonstrated the absence of cytotoxic effects, which in the case of receptor’s agonists are generally associated with off-target effects.   

The authors labelled the western blot image quantification values as fold-change. These are not fold-change values but relative intensity values with arbitrary units. Please change the y-axis labels appropriately.

A: The Reviewer’s observation is important. We have modified the Y-axis label accordingly to the Referee’s suggestion.

How did the authors fix the SUP-T1 cells on the slides for immunofluorescence analysis as these are non-adherent? The provided experimental detail is not convincing.

A: The Reviewer’s comments are very appropriate. Since SUP-T1 are non-adherent cells, we used an immunofluorescence protocol which included a fixation step based on the protocols generally used for metaphase spread preparation. Particularly, cells were collected and resuspended in a fixative solution (methanol and acetic acid, 3:1) for 20 minutes at -20°C. Finally, the cell suspension was dropped on microscope slides pre-treated with fixative solution and air-dried. After these passages cells were attached to the coverslips. These important technical details have been introduced into the Material and Methods Section.

STAT3 phosphorylation site was not mentioned in the method section. Please provide the detail. The authors could have provided the catalog number for each of the antibodies.

A: The Reviewer’s suggestions are again correct. Thus, we have introduced the STAT3 information in the Method Section together with all the catalog numbers of the employed antibodies.

Expansion of the abbreviation GPI assay was not provided (L247).

A: Thanks for the comment. We have explained the GPI abbreviation.

The supplementary table 1 was mentioned in the manuscript but not provided.

A: We thanks the Reviewer for his/her comment. We have introduced the requested Supplementary Table 1 in the relative section.

The authors have to modify the y-axis labels for supplementary figure 1A and 1B to identify the cell cycle phase they refer to. The cell-cycle analysis presented in this has “recovery” data group. Does this is an untreated group?

A: Thanks to the Reviewer for the keen observation. We have modified the Y-axis labels, accordingly. To better clarify the question, the Recovery (R) sample is an additional control in which cells were synchronized with Colcemid (as the treated samples) and, after 24 hours, the medium was replaced with a fresh one to remove the cell cycle blockage. This sample was used to determine the rescue of the cell cycle after Colcemid treatment and in absence of GEs. 

Reviewer 2 Report

Interesting paper, that could stimulate  clinical studies to identify the biological basis of symptomatology spectrum of celiac disease. Line 224, where is 9C in fig 9 ?

Author Response

REVIEWER 2 ANSWER

Interesting paper, that could stimulate  clinical studies to identify the biological basis of symptomatology spectrum of celiac disease. Line 224, where is 9C in fig 9 ?

A: We thanks the Reviewer for his/her comments, particularly for the encouraging perspective of next studies within celiac disease.

In the revised version of the manuscript, we have modified the underlined point at line 224, accordingly.